# LLM2Labels: Zero-shot dataset summarizing and labeling using foundational LLM models

## Abstract

We introduce the LLM2Labels framework for systematically generating a label vocabulary tailored to image segmentation within a comprehensive image dataset. This framework leverages the capabilities of Visual Language Models (VLMs) and Large Language Models (LLMs). Our methodology unfolds in two distinct stages. Firstly, we perform per-image processing, encompassing the Image Label Proposal and Filtering stage, comprising the Label Proposal Module (LPM) and the Label Filtering Module (LFM). In this stage, LPM employs VLMs to suggest candidate labels for each image, with consideration for the context of the task at hand. Subsequently, the suggested labels undergo a rigorous filtering process in the LFM, guided by a predetermined filtering strategy. Secondly, the Logical Grouping stage leverages well-established LLMs, notably Llama2, to empower the logical categorization of the meticulously filtered candidate labels. This categorization process resembles the organization of labels into coherent groups, akin to WordNet synonym sets. We assess the effectiveness of our framework on segmentation datasets, with a primary focus on ground truth segmentation labels within a closed-set scenario, while also revisiting the open-set evaluation. Notably, this research pioneers a novel application of VLMs and LLMs for zero-shot vocabulary discovery without manual annotators or experts. Our results reveal performance levels rival trained close-set multi-label classification while surpassing naive zero-shot models. This work signifies a pioneering leap in harnessing advanced language models for vocabulary generation in computer vision. Beyond its immediate applications in vocabulary creation for image segmentation, it promises to substantially benefit image analysis and research across the field.

## 1 Introduction

Establishing a coherent and effective vocabulary is paramount in image analysis and computer vision research. Image segmentation (Kirillov et al., 2023; Xu et al., 2022a; Minaee et al., 2021a; Chen et al., 2018; Ghosh et al., 2019), a widely used technique in computer vision and various domain-specific applications requiring precise semantic labels, underscores the need for high-quality descriptive labels. This study aims to craft a comprehensive set of descriptive labels tailored specifically for image segmentation within a given dataset. It aims to generate a vocabulary that aptly describes the types of regions occurring in the dataset. While, the essence of this challenge at the image level resembles that of multi-label classification (tagging) (Herrera et al., 2016), which is a well-established challenge (Ben-Baruch et al., 2022; Liu et al., 2021; Xu et al., 2022b), it distinguishes itself by the zero-shot methodology for vocabulary construction and the specific use for image segmentation.

The problem of finding the correct vocabulary is vital for creating labels for existing and new datasets. Annotating datasets at scale is complex, time-consuming, and expensive as more classes are added (Russakovsky et al., 2015; Gupta et al., 2019). Finding the labels that fit the dataset for a specific task (in our case, segmentation) is the first step in the pipeline for dataset annotation. Moreover, the proposed vocabularies of the dataset can be used downstream on existing open vocabulary methods (Xu et al., 2023; Ding et al., 2022), thus obtaining initial segmentation maps for the critical concepts in the dataset. Automatic vocabulary computation for image segmentation promises to have a high impact by alleviating the labor-intensive and costly vocabulary creation, ultimately accelerating the dataset annotation process while maintaining label quality and dataset coherence.

In addition, combined with a zero-shot segmentation method, one is able to fully automatically segment a dataset without any human intervention.

Several approaches have been explored to create a vocabulary for image segmentation, each with advantages and challenges. The first approach, manual labeling, involves human annotators meticulously assigning labels to images by iteratively observing and refining them (Gupta et al., 2019; Everingham et al., 2015; Lin et al., 2014). While this method ensures high precision in labeling, it is labor-intensive, time-consuming, and cost-prohibitive, especially for extensive datasets, and introduces potential human subjectivity and errors.

The second strategy frequently employed involves the utilization of pre-defined vocabularies, including well-known resources such as WordNet (Miller, 1995), ImageNet (Russakovsky et al., 2015), and specialized datasets like COCO (Lin et al., 2014) and Pascal VOC (Everingham & Winn, 2012). These established vocabularies offer structured labels as a foundation for labeling and downstream tasks. However, it's essential to acknowledge the inherent limitations associated with this approach. While these vocabularies provide stability and structure, they often lack the adaptability to encompass new or evolving concepts, primarily functioning as standardized benchmarks. Moreover, the use of large vocabularies containing more than multiple thousand words can pose significant computational challenges within the current standard approaches to open vocabulary segmentation (OVS) (Ghiasi et al., 2022; Xu et al., 2023). Specifically, managing large vocabularies presents practical difficulties for downstream segmentation algorithms that rely on image-proposed region similarities (Radford et al., 2021a; Xu et al., 2022a).

Thirdly, Tag2Text (Huang et al., 2023) employed a hybrid approach named category creation, combining automated methods with manual filtering to amass a collection of 5 thousand categories. These categories were extracted from diverse datasets, including CC-3M (Sharma et al., 2018), COCO (Lin et al., 2014), Visual Genome (Krishna et al., 2017), and SBU (Ordonez et al., 2011). Notably, the first two datasets are human-annotated, while the latter are web-based resources. The process of tag extraction relied on a semantic parser (Wu et al., 2019) to discern nouns within the aforementioned datasets. However, it's worth noting that the resulting tag categories lacked a hierarchical structure or synonymous groupings (synsets Miller (1995)) in addition to the required ground truth captions. While not directly tailored for image segmentation, they were harnessed for multi-label purposes, encompassing objects, scenes, attributes, and actions. In the end, the vocabulary is still fixed.

In contrast to vocabulary discovery for a complete dataset, multiple approaches to obtaining multi-labels from a single image can be combined based on frequency to form a vocabulary. Approaching the extraction of tags per image with predefined vocabulary-supervised models. This would include the trained Tag2Text, which can be combined with the same frequency aggregation. Recent works by MKT (He et al., 2023) have shown promising results, achieving open vocabulary performance with 5% to 8% unseen classes. Commercial API services, including those offered by Apple, Microsoft, and Google (Apple; Microsoft; Google), are available for addressing the per-image multi-label classification problem.

We propose a framework called LLM2Labels for a zero-shot label proposal in 2 stages. Firstly, we input images into the Label Proposal Module (LPM) and apply the Filtering Module for each image. The second stage consists of basic filtering of the outputs and grouping this to a dataset using LLM as a reasoning engine for grouping similar concepts. This framework uses language-aligned models such as InstructBLIP (Liu et al., 2023) for the LPM. Similarly, we use existing LLM, such as LLAMA2 (Touvron et al., 2023), to group the semantically similar concepts into synsets. Those labels can be used in existing downstream open vocabulary models (e.g., ODISE, OpenSeg Xu et al. (2023); Lüddecke & Ecker (2021) and similar works discussed in survey Zhu & Chen (2023)).

More specifically, our framework works for a given set of images. We find possible candidate labels per image and filter them according to a filtering strategy. In our work, we extract existing probabilities of the VL-LLM to filter out the labels. Afterward, we use those label candidates before feeding the possible tags to an LLM to group and summarize into logical groups. Segmentation was chosen as the specific task for evaluation purposes as it encompasses multiple semantic meaningful tags compared to the task of object detection. We evaluate produced labels on different segmentation datasets against ground truth segmentation labels, which are assumed to be closed-set.

Our contributions are the following:

1. To our current understanding, LLM2Labels represents the first work to investigate the utilization of Large Language Models (LLMs) in the context of generating vocabularies for image datasets.

2. We propose a novel framework for proposing labels, filtering them, and summarizing using LLMs for an open-set vocabulary generation

3. We outperform existing close-set multi-label classification and naive zero-shot models evaluated on segmentation datasets.

## 2 RELATED WORK

**Dataset labels.** Most works use existing datasets that have a limited number of tags of interest. Training previous models on a small number of labels, compared to natural language expressivity, such as 20 classes from PASCAL VOC 2012 dataset (Jing & Tian (2020)), Pascal Context (Mottaghi et al., 2014) with a subset of 59 commonly used (extension of PASCAL VOC with more than 400 classes), MS-COCO (Lin et al., 2014) with 133 total classes (thing, stuff and merged) and Cityscapes (Garcia-Garcia et al. (2017)) with 30 categories. The dataset that is closest to free-form natural language expression is ADE20k (Zhou et al. (2017)) with a common subset SceneParse150 Zhou et al. (2019) of 150 classes (from the 3,688 classes with WordNet definition and hierarchy). The ADE20k structure is the closest label structure we want to achieve with our framework. LVIS Gupta et al. (2019) dataset has 1000 object categories used, for instance segmentation. Those labels are loosely based on WordNet taxonomy to compute similarities and synonyms.

**Image segmentation.** Image segmentation was traditionally approached using specialized datasets for the task (), and more recent approaches using general image text datasets. Because of the trend to go towards open vocabulary segmentation (OVS) we have a lot of work in this direction. Segmentation algorithms can be categorizes as either semantic, instance, or panoptic segmentation (combines instance and semantic). Different model architectures are proposed to achieve each of those levels, including DeepLabv3 Chen et al. (2017; 2018), GAN-based approaches Luc et al. (2016); Choi et al. (2019), transformer-based approaches Li et al. (2023b), and diffusion-based approaches Amit et al. (2021). For a more holistic overview of deep learning models for Image Segmentation, check Minaee et al. (2021b). There are also interactive techniques which leverage deep learning ... TODO Laradji et al. (2020); Kirillov et al. (2023)

**Zero shot methods** Recent works provide a new paradigm in image segmentation. We want to label existing class labels of a dataset and the wild images given a textual label or image exemplar. Such works usually use some vision understanding models (mentioned in section 1). One such work, CLIPSeg Lüddecke & Ecker (2021), uses CLIP hidden states alongside clip embedding to train a simple decoder with a segmentation mask as output. Similarly, ViewCo Ren et al. (2023) uses textual supervision of CLIP embeddings but enforces multi-View Consistent learning of different crops to the same text input. Concurrent works like ODESI use a similar approach Xu et al. (2023), which uses the diffusion UNet model's features instead of CLIP (with the textual caption as conditioning), in addition to text encoder of category and image caption labels (nouns from the caption).

**Vision and language models.** Models like CLIP Radford et al. (2021b) have bridged the NLP and computer vision community. Tasks have expanded from Image-text pretraining (Image captioning), visual classification question answering (VQA), Video Question Answering (VideoQA), and others (Li et al. (2021; 2022)). Models like VisualGPT Chen et al. (2022) show the multi-modal application of Pretrained Language models (PLMs) for vision tasks.

**Concurrent works** in the field, such as SegGPT Wang et al. (2023) use an in-context approach, given an image example of the segmentation, to produce a segmentation invariant to a dataset's labels following Wang et al. (2022). While this approach generates accurate panoptic segmentations, it still relies on the user's proposed regions (and naming of such). However, this is a small problem given a dataset's existing context (known labels). LLaVA, Large Language and Vision Assistant, is another approach that improves zero-shot capabilities of LLMs on new tasks in the multimodal field.

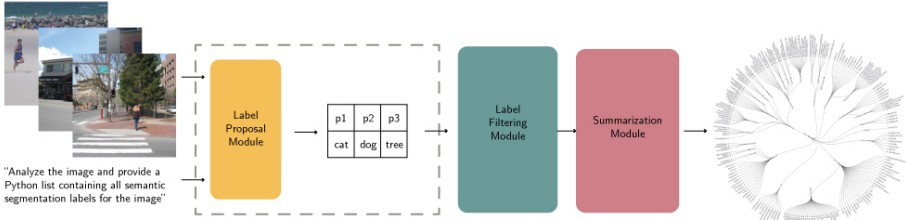

Figure 1: Overview of the general pipeline of our LLM2Label framework, which consists of 2 stages. The first stage generates a set of candidate labels $\mathcal{L}$, and the second stage is used to construct the vocabulary given those predicted candidate labels.

# 3 METHODOLOGY

## 3.1 PROBLEM DEFINITION

Given a set of images $\mathcal{I}_\lceil = \{\boldsymbol{b}i_1 \dots \boldsymbol{b}i_n\}$, we produce a vocabulary $\mathcal{V}$ using an LLM which can be further used to perform various downstream tasks. In our work, we focus on zero-shot segmentation. We impose some constraints to keep the automatic and generally applicable, mainly that no human annotators or experts are involved in the end-to-end pipeline. This is in contrast to the approach taken by manual vocabulary creation. Moreover, even though we evaluate against a limited number of classes in the ground truth label annotations, we make no assumptions about the number of labels in each image or the dataset itself.

## 3.2 METHOD OVERVIEW

Our work aims to construct a vocabulary in an unsupervised, zero-shot fashion for image datasets using Vision Language LLMs. The overall two-stage framework is depicted on fig. 1. We discuss the strategy and information used to extract labels section 3.3 and the method to summarize tags in section 3.4.

## 3.3 LABEL PROPOSAL STAGE

In the first step, given a dataset of images $\mathcal{I}_\lceil = \{\boldsymbol{b}i_1 \dots \boldsymbol{b}i_n\}$, where $i_1 \dots i_n$ are the indices for the $n$ images in a dataset, we aim to derive a set of candidate labels $\mathcal{L} = \{\boldsymbol{b}l_{j_1} \dots \boldsymbol{b}l_{j_k}\}$ where $j_1 \dots j_k$ are the indices of represent $k$ non-unique proposals for the $i$-th image. In cases where $n$ equals 1, $\mathcal{I}_\lceil$ is equivalent to $\mathcal{I}$, denoting a solitary image, thereby conforming to the conventional paradigm of a multi-label problem in this context.

Extracting the labels from an image $I$ or a dataset $\mathcal{L}$ using LLMs is not a trivial task, as LLMs are prone to hallucinations (Borji, 2023; OpenAI, 2023; Bang et al., 2023), which also translates to vision-language models as shown on object detection (Li et al., 2023c). We focus on three goals for the first stage. Firstly, have open-vocabulary suggestions for labels more specific to the dataset for intra-class (e.g., 'wooden bench' in addition to 'bench') and specialization (e.g., 'nightstand' in addition to 'furniture'). The second goal is to use existing information of models to provide more control over the label suggestion process. Lastly, an accurate per-label proposal strategy is needed to achieve a reasonable $\mathcal{L}$ for the second stage.

We adopted an additive approach that utilizes vision-language models without relying on predefined categories, such as WordNet or ImageNet (Miller, 1995; Russakovsky et al., 2015) to achieve the first goal. This strategic choice was made to ensure semantic flexibility and facilitate the creation of an open vocabulary tailored to the nuances of a specific dataset. The vision-language models used are known for their ability to represent each word with a high degree of granularity (Kudo & Richardson, 2018).

We achieve more control over suggested labels by the model and remove suggested hallucinations by extracting for each token $y_i$ the corresponding probabilities $p(y_i)$ in eq. (1) for each candidate

label from **L**. More specifically, we sample with nucleus sampling (Welleck et al., 2019) from scaled probabilities of the subset $V^{(p)}$ that satisfied eq. (2). The probabilities are normalized across the vocabulary of the selected vision-language model. The probability $P(y_1...y_n|\boldsymbol{x})$ is calculated on context $\boldsymbol{x}$ (i.e., prompt) and current token $y_i$. Later, in section 4, we investigate the trade-off of different filtering methods given conditioned probabilities with image tokens in $\boldsymbol{x}$.

$$P(y_{1:n}|\boldsymbol{x}) = \prod_{i=1}^{n} p(y_i|y_{1:i-1}, \boldsymbol{x}) = \prod_{i=1}^{n} p(y_i|y_{<i}, \boldsymbol{x}) \tag{1}$$

$$\sum_{y_i \in V^{(p)}} P(y_i|y_{<i}) \geq p. \tag{2}$$

### 3.3.1 FILTERING

The next step of the method is to refine the candidate labels, improving the per-image label proposals. As it turns out, the set union of the proposed labels in section 3.3 is insufficient for the model to identify all the labels of interest in an image due to the region's size, location near edges, and ground truth synonym definition mismatch (e.g., predicted 'motorcycle' compared to 'motorbike'). The first two can be tackled with an increase in label proposals, to the detriment of precision. In addition, we ablate different design choices to observe the representation influence on precision and recall for our label choices. As for the latter, we mainly follow the label synonym choice of ground truth from ODISE (Xu et al., 2023).

With the previous trade-off for recall, we explore multiple design choices for the filtering stage to reduce the number of false positives as a self-correction step. Namely, we explore self-correction by the model (e.g., re-prompting the model with a prior from the produced labels), non-overlapping image griding, multiple ways of representing the probability of tokens, occurrence of labels, and combining the probability with occurrences. We deal with various levels of combining the scores, mainly because we do not have a fixed output vector of probabilities like supervised methods such as Huang et al. (2023). Even with a single image, if prompted for a caption, we can have multiple responses of the same label, which is even more exaggerated in a multi-sampling setting. Namely, the inner stage could be represented as seen on eq. (3), but the process is repeated on the sample level and, finally, the dataset level for the same proposed labels (more information appendix D).

$$P(b_{l_i}) = P(b_{l_{1...k}}) = \begin{cases} \frac{1}{k}\sum_{i=1}^{k} b_{l_i}, & \text{if } f = \text{avg} \\ \prod_{i=1}^{k} b_{l_i}, & \text{if } f = \text{product} \\ \min(b_{l_i}), & \text{if } f = \text{min} \\ \max(b_{l_i}), & \text{if } f = \text{max} \end{cases} \tag{3}$$

### 3.4 VOCABULARY CREATION

We can see that most initial vocabularies were a small number of classes, with shallow or no hierarchy. We aim to investigate the automatic creation of such vocabularies for a given dataset because of the limited token size available in most large language models. We use LLAMA2 from Touvron et al. (2023), which has a context size of 4k tokens, including sub-tokens and other filler tokens. We aim to filter out most proposed labels before arriving at this stage.

In this stage, given the refined proposals from the filtering stage, which are ranked based on methods estimate of the label presence in the whole dataset, we construct a vocabulary using an LLM to a logical group (synsets), refine synonyms, and construct an image vocabulary **V**. In addition, we also derive the hierarchy represented as a set of trees, $\mathcal{H} = \{\boldsymbol{b}h_{j_1} \ldots \boldsymbol{b}h_{j_m}\}$ where $j_1 \ldots j_m$ represents the indices of $m$ such trees. Here, the root of each tree represents a dominant label in an image or a dataset.

## 4 EXPERIMENTS

This section will start with a brief look at the datasets used, methods used for comparison, followed by the metrics used to evaluate such a task, and finally, explore the results achieved for both label

Table 1: Comparison with **seen** ground truth vocabulary ranking of different methods on ADE20K under 2 configurations (A-150 and A-847). Precision and recall are reported at the K={150, 1000} for A150 and A847, respectively, and AUC(PR@K) for $K \in [1, 5000]$. All configurations use a threshold of 0.68.

| Methods | Supervised | A-150 | | | A-847 | | |
|---|---|---|---|---|---|---|---|
| | | P | R | AUC | P | R | AUC |
| BLIP2 + SP Wu et al. (2019) | ✗ | 34.67 | 39.10 | 27.85 | 15.47 | 56.64 | 10.09 |
| BLIP2 VQA (Li et al., 2023a) | ✗ | 13.06 | 41.67 | 41.39 | 18.10 | 66.8 | 34.55 |
| Tag2Text (Huang et al., 2023) | ✓ | 45.71 | 42.67 | 32.69 | **22.63** | 65.23 | 24.46 |
| LLM2Label(Ours) | ✗ | **54.09** | **48.50** | **46.23** | 19.27 | **71.19** | **39.00** |

proposal stages (sections 3.3 and 3.3.1) and vocabulary creation stage (section 3.4). All experiments were done on an A100 80GB GPU. We don't train the models but rather prompt a series of models to achieve our desired outcome without extra training. Additional details of the exact parameters used will be provided in appendix D.

**Datasets.** We use ADE20K (Zhou et al., 2017) under two configurations, A-150 and A-847, where the number depicts the number of classes in the vocabulary, where class label synonyms used for the evaluation were obtained from Xu et al. (2023). In most of the experiments, which are 2k images, under the A-857 we actually come across 256 out of the 847 classes.

**Methods.** We compare our approach with supervised and zero-shot methods. **Tag2Text** (Huang et al., 2023) is supervised and is trained on 14M images for tagging, including the COCO dataset (Lin et al., 2014). We also compare against two zero shot methods that are unsupervised. **BLIP2 + SP** is a combination of image captioning using BLIP2 (Li et al., 2023a) and a semantic parser (Wu et al., 2019). **BLIP2 VQA** by Li et al. (2023a) uses visual question answering. We use occurrence aggregation on the vocabulary level for other baselines, normalizing occurrences per image. Specifically for Tag2Text, we apply the best threshold reported by Huang et al. (2023) while generating per-image proposals. In our method, we represent each label as a product of occurrence and extracted probabilities mentioned in section 3.3. This has been shown to increase per-image proposal accuracy over different thresholds with the added benefit of fewer incorrect candidate labels, described in appendix B. We apply different ways of representation based on extracted probabilities. While our focus is not single image evaluation, we provide a comparison in appendix A, as the metric suffers from an open vocabulary situation.

**Metrics and Evaluation Protocol.** In the first stage, our primary objective is to generate a ranked list of predictions, which will be the top predictions for subsequent stages. Our main objective is to find a method that consistently ranks the top K with higher precision and accuracy. Results were reported using a threshold of 0.68 for Tag2text. Likewise, we report results for zero-shot methods using the same threshold unless otherwise stated but with a slight change. Changing the meaning slightly, a threshold of 0.68 is a percentile of low-ranked labels discarded per image for the zero-shot methods. Mainly to combat the increase of sampling influence of precision. We apply the same logic of selection of top K in Huang et al. (2023), which in turn guides to image retrieval metrics. More specifically, we pose the problem as an information retrieval task for the correct labels and use adequate precision, recall at K metric, and the AUC(PR@K). Synonyms were used for matching, but we only counted them once for a single class score, regardless of the number of synonyms found because of the disparity of synonyms for classes.

We report each method's performance for R@K, P@K, and AUC(PR@K) in 2 scenarios. Firstly, we compare the methods on the ground truth labels of the seen labels. Then, we compare against the full vocabulary (out of which we report unseen %) to showcase the generalization of the recognition. Likewise, all the reported results are an average of 4 runs, with different seeds with the same filtering strategy and a subset of images unless otherwise stated. Exact parameters, prompts, and other information can be found in supplementary (appendix D). The threshold used for comparison is solely chosen to be the same as Huang et al. (2023). They report the best results of Tag2Text at a threshold of 0.68, which are used for comparison, with minor modifications for all zero-shot methods using percentile-based filtering compared to the fixed probability of a supervised method.

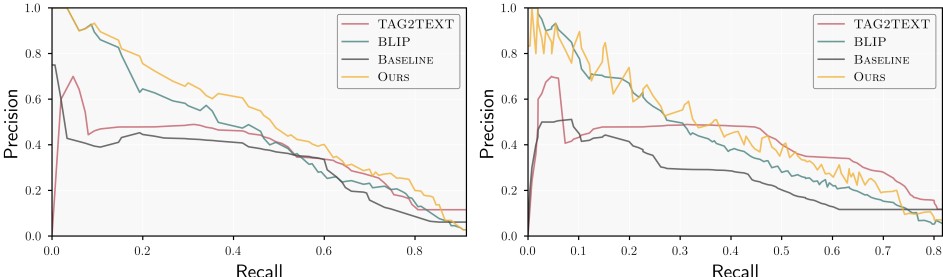

Figure 2: Precision-recall curves for the results in Table 1. To the left, we show results for A-150 and on the right, we show results for A-847.

## 4.1 Comparison

We report the chosen metrics in Table 1, where we can observe that LLM2Label outperforms all other zero shot baselines while being only outperformed in precision for A847 by the supervised method Tag2Text. Tag2Text was already exposed to the majority of the classes during its supervised training. Therefore, it is surprising that we can surpass Tag2Text in almost all metrics with our zero-shot method. A better understanding of the behavior of different methods comes from studying the precision-recall curves in Section 4. We can see that overall our method and BLIP2 VQA have the best performance, especially in settings where we aim for high precision. Tag2Text's performance is better at the tail end of the curves where recall is very high. Here the advantage of supervised training shows, that Tag2Text has been exposed to most of the labels during supervised training. Overall, we would mainly emphasize the metric AUC to argue that our method is the overall best, even beating out the supervised method. More details concerning the Tag2Text supervised method are given in Appendix C. We also show qualitative results on example images for illustration in Figure 4.

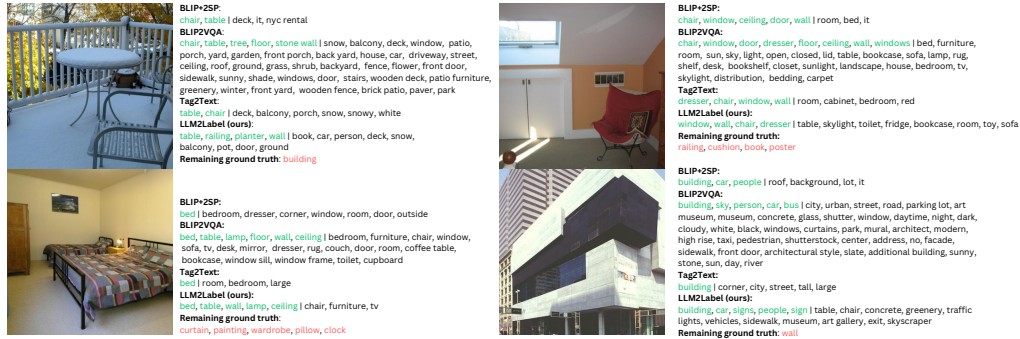

Figure 3: Example results of LLM2Label and competing methods in the first stage. We use thresholding based on the $68th$ percentile for the zero-shot methods and a $0.68$ probability threshold for Tag2Text.

## 4.2 Ablation Study

In this section, we explore different design choices explored in this work. We compare the following variants. Base: the full image is processed eight times. For each time the image is processed a (possibly different) set of labels is sampled. Labels are filtered with a percentile-based threshold of $68\%$ SG: the image is divided into a grid of $2 \times 2$ images and each of the four sub-images is processed two times. SGC: the full image is processed once and each of the $2 \times 2$ sub-images from the previous variant are processed once. TH: instead of using a percentile threshold a fixed threshold of $0.68$ is used for the probability. NONE: no thresholding is employed. BOTH: we employ a percentile-based threshold of $68\%$ and a fixed probability threshold of $0.68$. The results are shown in Table 2. We can see that both percentile and fixed thresholding for the per-image labels create a lot fewer label candidates. This is illustrated by the metric $\#L$ that reports the total number of labels discovered.

Table 2: This table explores various design choices and configurations. Specifically, 'Base' represents a holistic image perspective, 'SG' indicates sub-gridding, and 'SGC' combines both approaches. The subsequent section evaluates different threshold options on 'Base', where our default is percentile-based filtering, 'th' representing the standard thresholding technique in supervised models, and 'none' indicating the absence of such technique. All reported configurations are based on the average of four runs, each employing a threshold value of 0.68, except for the 'none' scenario. We also indicate the number of proposed labels as #L (at $K = 5k$)

| Methods | A-150 | | | | A-847 | | | |
|---|---|---|---|---|---|---|---|---|
| | P | R | AUC | #L | P | R | AUC | #L |
| LLM2Label$_{BASE}$ | 54.02 | 49.33 | 41.94 | 1914 | 29.19 | 60.64 | 35.02 | 1174 |
| LLM2Label$_{SG}$ | 53.47 | 47.50 | 46.72 | 435 | 30.11 | 57.35 | 44.58 | 618 |
| LLM2Label$_{SGC}$ | 52.81 | 47.67 | 44.32 | 320 | 37.81 | 47.27 | 30.32 | 321 |
| LLM2Label$_{TH}$ | 56.64 | 50.50 | 47.45 | 855 | 29.11 | 60.64 | 35.08 | 533 |
| LLM2Label$_{NONE}$ | 52.93 | 46.67 | 47.26 | 4881 | 19.07 | 69.53 | 39.17 | 3386 |
| LLM2Label$_{BOTH}$ | 56.64 | 50.50 | 47.46 | 858 | 29.19 | 60.64 | 35.02 | 547 |

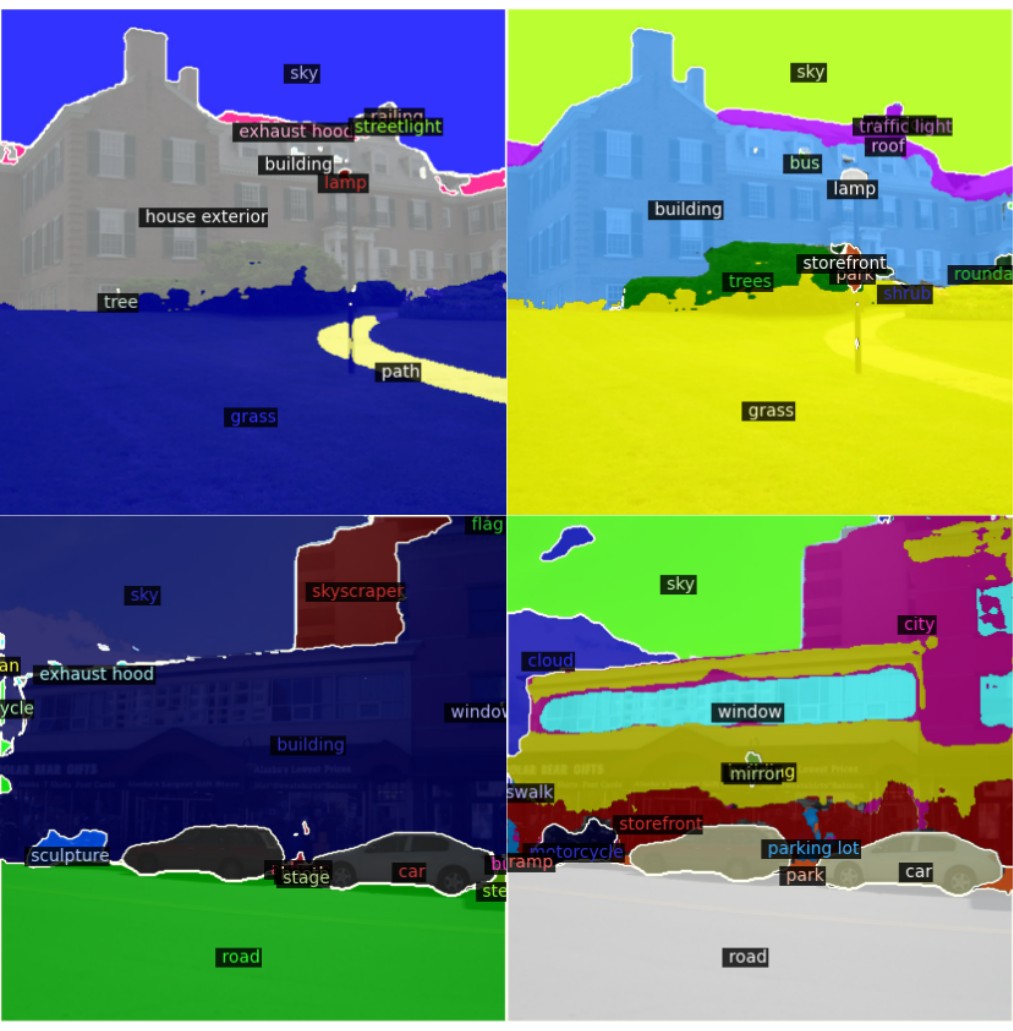

Figure 4: Example zero shot results of SegFormer on default vocabulary left column, and right column, results from the same model with our vocabulary.

The filtering removes label candidates already at the per-image processing stage so that less likely labels will not be considered. Otherwise, most results are pretty comparable, trading off precision and recall in different ways.

### 4.3 SEGMENTATION RESULTS

We show qualitative results of segmented images in Figure 4. We use SegFormer to obtain zero-shot segmentation results. We demonstrate that using our vocabulary can bring a more fine-grained segmentation compared to using the default vocabulary of SegFormer.

## 5 CONCLUSION

We proposed the LLM2Labels framework for systematically generating a label vocabulary tailored to image segmentation for an image dataset. We employ a combination of per-image and global per-dataset processing. Our results demonstrate that LLM2Labels can not only beat other zero-shot methods, but also the supervised method Tag2Text in quantitative metrics. In future work, we would like to extend our framework to videos and 3D object datasets.

## 6 LIMITATIONS

The limitation of this work stems from the limitations of the reasoning abilities of vision-language models and the limitations of the visual query abilities of existing models. This includes the problem of the models not producing a desired output of the input image, including but not limited to repeating region suggestions, synonyms, or hallucinations of things that do not exist on the scene. Our method can mitigate these problems, but the critical ones are no or majority faulty proposals. Moreover, while in the initial stages of exploration, we envisioned a specialized system. Examples include subspecies of fish such as Ulucan et al. (2020), microscopic images, and other non-standard vision domains. We concluded that the models can sometimes guess (within the top five or a few runs). This needed to be improved for such a specialized dataset understanding, especially the level for the medical image domain.

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

Table 3: Performance comparison of **image** level (multi-label classification of labels) on ADE20K under 2 configurations (A-150 and A-847). Reported mAP at 0.68 thresholds and AUC under Precision-Recall curve across all thresholds (close set evaluation).

| Methods | Supervised | A-150 | | A-847 | |
|---|---|---|---|---|---|
| | | mAP | AUC | mAP | AUC |
| BLIP2 + SP (Wu et al., 2019) | ✗ | 15.05 | 7.76 | 28.99 | 0.03 |
| BLIP2 VQA (Li et al., 2023a) | ✗ | 0.14 | 7.76 | 28.99 | 0.34 |
| Tag2Text (Huang et al., 2023) | ✓ | 19.3 | 29.3 | 28.27 | 5.40 |
| LLM2Label (ours) | ✗ | 19.62 | 19.56 | 28.99 | 0.43 |
| LLM2Label$_{occ}$ (ours) | ✗ | 21.97 | 14.55 | 28.99 | 0.38 |

Table 4: Results of using probability on LLM2Label number of label reduction while keeping performance. This effect is mainly noticeable on K higher than one thousand, while the same behavior is below 1k. The results are an average of 4 runs at $K = 5000$, rounded to the nearest number. Both scenarios use the same percentile filtering of 0.68 thresholds.

| Method | Number of FP in vocab | |
|---|---|---|
| | A-150 | A-847 |
| LLM2Label (ours) | 1353 | 790 |
| LLM2Label$_{occ}$ (ours) | 3599 | 2356 |

## A  IMAGE LEVEL COMPARISON

While the primary goal of our paper is to construct a good vocabulary, we explore the per-image performance in a close set setting, where we need to count correct labels only seen in the images. This is per image level equivalent of the table 1 in section 4.1. Although the model performs better on A-150 with only occurrences, we observed a performance gain in the whole threshold region. This makes it an easier choice when the performance is better when you increase the number of classes, even though the increase is slightly to 256 observable classes compared to the complete setting. Note that we report numbers only on mapped classes to the target dataset by close-set evaluation, excluding any label not mapped to the synonym set of the ground truth. This is not the case in section 4.1, where we perform the same mapping but report unmapped classes as misses.

## B  USE OF PROBABILITY

As observed in table 3, we can see that introducing probabilities increases precision on threshold range while showcasing improvements in filtering ability table 4. We assume this has to be due to probabilities of tokens already being conditioned on the image itself.

## C  COMPARISON DETAILS FOR TAG2TEXT

Although Tag2text (Huang et al., 2023) is not directly trained on finding semantic regions of the image, authors do indicate a high-label class overlap in the final training set, and the procedure for semantically meaningful objects does occur more in the descriptions. This is mainly because it was trained with COCO (Lin et al., 2014) (among others) with authors of Huang et al. (2023) confirming overlap of 73 and 358 classes from COCO (Lin et al., 2014) and ImageNet (Russakovsky et al., 2015), respectively. We did ensure that Tag2Text has 132 overlap classes with the A-150 configuration of ADE20K (Zhou et al., 2017). Tag2Text initial overlap is 192 classes before the mapping process, with different synonyms and even repetitions of the same label class. During testing, we mainly take the maximum of the two overlapping classes.

## D    ADDITIONAL DETAILS

All experiments done for our method were using InstructBLIP, with nucleus sampling of $p = 0.9$, with prompts "Analyze the image and provide a Python list containing all semantic segmentation labels for the image." and similar prompts for the first module and different variations of the following prompt: "Use the list below, to construct of the hierarchy of dataset labels, and produce a python dictionary, such that the hierarchy describes the labels given describes the dataset, strictly only using the ones provided:List of proposed labels". We use a maximum length of 100 new tokens per sample, with up to 8 times sampling per image (depending on configuration). The length penalty was disabled to closer to probabilities, but the repetition penalty was fixed to 2.0. Temperature was fixed at 0.3 for the majority of experiments that use sampling.

