# OpenReview forum: "LLM2Labels: Zero-shot dataset summarizing and labeling using foundational LLM models"
_ICLR.cc/2024/Conference — ICLR 2024 Conference Withdrawn Submission_

### Official Review · Reviewer_UakG · 2023-10-21

**Soundness:** 1 poor
**Presentation:** 1 poor
**Contribution:** 1 poor
**Rating:** 1
**Confidence:** 5

**Summary:**

This paper proposes to use LLMs and vison-language models to label a dataset automatically. The procedure includes the vocabulary proposal, filtering, and summarizing.

**Strengths:**

1. This paper might be the first one to systematically analyze how to label a dataset automatically using large pre-trained models.
2. The authors incorporate different models in different stages to benefit from their distinguished advantages.

**Weaknesses:**

This paper is obviously below the requirements of a top conference for the following weaknesses.

Formats
1. This paper does not use the template for reviewing, the submitted version does not have line numbers.
2. Figure 1 is not clear enough.
3. Many typos like the use of "" in section 3.3.

Methods
1. The proposed pipeline is straightforward, I do not see any technical contributions.
2. Models like BLIP2 and LLAMA might not be the best choices, I believe MLLM models like LlaVA, mini-GPT4, and KOSMOS-2 would give better performance.
3. The authors pick the segmentation dataset as their demos. However, how to get the mask proposals and how to align each mask with labels are not solved, which is a key problem.

**Questions:**

See the weaknesses.

---

### Official Review · Reviewer_Het9 · 2023-10-24

**Soundness:** 3 good
**Presentation:** 2 fair
**Contribution:** 2 fair
**Rating:** 5
**Confidence:** 3

**Summary:**

This paper considers to generate a label vocabulary tailored to image segmentation within a comprehensive image dataset and utilize LLM to empower the logical categorization of the meticulously filtered candidate labels

**Strengths:**

The paper contains two stage. In First stage, LLM2Labels propose Label Proposal Module (LPM) and the Label Filtering Module (LFM). In second stage, LLM2Labels utilize LLM to empower the logical categorization of the meticulously filtered candidate labels.

**Weaknesses:**

1. The novelty of this paper is limited. Many recent works research on generating vocabularies for images. (including fine-grained objects). (e.g., RAM, SegGPT and open-vocabulary detectors.). Authors do not clearly introduce the details of this paper and I do not see the obvious contributions.

2. Formula in Sec 3.3.1 is not clear. What is the meaning of each symbol. It appears to be one of only three formulas in the paper.

**Questions:**

As mentioned above in the weakness.

---

### Official Review · Reviewer_afgG · 2023-11-22

**Soundness:** 2 fair
**Presentation:** 2 fair
**Contribution:** 2 fair
**Rating:** 5
**Confidence:** 5

**Summary:**

This paper proposed LLM2Lables to handle closed-set image segmentation task and revisit open-set scenario as well. LLM2Labels contains two stages, i.e., per-image processing for image labels achieving and logical categorization of filtered candidate labels.

**Strengths:**

The task handled by this paper has been important in recent years.

The overall writing and organization are clear.

**Weaknesses:**

The motivation of two-stage processing is not novel, i.e., it has been explored by other LLM-based methods. The proposed LPM and LFM are also not well motivated and more in-depth analysis should be given.

This paper claims to use LLM and VLM, however, how to modulate these two sides are not well disscussed. This degrades the contribution.

This paper missed many relevant methods published in CVPR 2023, ICCV 2023, etc.

**Questions:**

See above

---

### Official Review · Reviewer_VpyB · 2023-11-26

**Soundness:** 2 fair
**Presentation:** 1 poor
**Contribution:** 1 poor
**Rating:** 3
**Confidence:** 4

**Summary:**

The paper introduces the "LLM2Labels" framework, designed for generating label vocabularies specifically for image segmentation tasks in extensive image datasets. The first stage involves per-image processing, where VLMs propose candidate labels for each image. The second stage uses LLMs, particularly Llama2, for logically categorizing these labels into coherent groups, similar to WordNet synonym sets. The framework is tested on segmentation datasets, showing promising results in ground truth segmentation labels, both in closed-set and open-set scenarios. It outperforms traditional zero-shot methods and even rivals trained close-set multi-label classification.

**Strengths:**

1. The framework's two-stage process, comprising the per-image processing (label proposal and filtering) and a grouping stage, is promising. This structured approach ensures that the labels generated are not only contextually relevant but also logically organized, akin to WordNet synonym sets.

2. Compared with some naiive baseline like BLIP and BLIP-VQA, this framework achieves better matching score compared to ground-truth label, showing a post-filtering LLM grounding is useful.

**Weaknesses:**

My major concerns are two-folded:

1. The implementation of the proposed framework is very unclear. Even after reading the paper for multiple iterations, I still didn't find which VLM is used for the per-image proposal stage; Also, how it is set to generate candidates, how many are generated, how they can be further filtered by LLM, are all very unclear to me. The authors shall include all implementation details (and better to include code) for easier re-implementation;

2. Generally I like the summarization module that organize some candidates entities by VLM into a structured taxonomy. However, the current implementation looks a naiive clustering, without iterative refinement or constrained by some grammar. There's very likely that LLM simply ignore some provided entities. Also, though in Fig1 also shows a hierarchical tree as output, I didn't see how the introduced approach can be used to construct such taxonomy. Probably the authors could provide more explanations here.

3. Why authors just use segmentation as evaluation task? Also how it can generalize to different datasets (with different requirement). I think it's very hard for BLIP model to generate correct candidates for different datasets as they are created with different purpose (say, a pathology dataset where all label are cell name).

Small typo: at the end of related work for image segmentation, it says: which leverage deep learning ... TODO

**Questions:**

How the approach is implemented and how it can generalize across datasets & tasks.

---

### Author Response · Authors · 2023-11-28
**Thank you for your review**

Thanks for your insightful feedback on our ICLR submission, especially insight gained from reviewer VpyB. After carefully considering it, we decided to withdraw the current submission to address the identified issues and deliver a more polished version of the work.